# Transformer Neural Networks in the Measurement of $t\bar{t}H$ Production in the $H \rightarrow b\bar{b}$ Decay Channel with ATLAS

**Chris Scheulen[1]⋆, on behalf of the ATLAS Collaboration**

**1** Université de Genève, Geneva, Switzerland

⋆ chris.scheulen@cern.ch

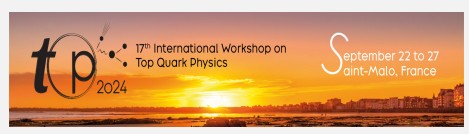

## Abstract

A measurement of Higgs boson production in association with a top quark pair in the bottom–anti-bottom Higgs boson decay channel and leptonic top final states is presented. The analysis uses $140\,\mathrm{fb}^{-1}$ of 13 TeV proton–proton collision data collected by the ATLAS detector at the Large Hadron Collider. A particular focus is placed on the role played by transformer neural networks in discriminating signal and background processes via multi-class discriminants and in reconstructing the Higgs boson transverse momentum. These powerful multi-variate analysis techniques significantly improve the analysis over a previous measurement using the same dataset.

## 1 Introduction

The Yukawa coupling of the top quark – the heaviest particle in the Standard Model – to the Higgs boson is suspected to play a critical role in electroweak symmetry breaking and the stability of the Higgs vacuum potential. While the coupling can be measured indirectly from the quantum loops underpinning the effective Higgs boson couplings to gluons and photons, its direct measurement is only feasible in processes involving the associated production of Higgs bosons and top quarks.

In this context, the ATLAS collaboration [1] at the Large Hadron Collider [2] recently published a measurement [3] of the associated production of a top quark pair with a Higgs boson ($t\bar{t}H$) in the bottom–anti-bottom Higgs boson decay channel ($H \rightarrow b\bar{b}$) with the full Run 2 dataset encompassing $140\,\mathrm{fb}^{-1}$ of proton–proton collision data collected between 2015 and 2018. $t\bar{t}H$ production has been discovered by both the ATLAS [4] and CMS [5] collaborations in 2018 in combination measurements of several Higgs boson decay channels. The $H \rightarrow b\bar{b}$ decay channel of $t\bar{t}H$ production – together called $t\bar{t}H(b\bar{b})$ – offers a large branching fraction of 58 % [6] and the possibility to fully reconstruct the Higgs boson four-momentum at the cost of a large irreducible background consisting of top quark pair production in association with

additional – mostly heavy-flavour – jets ($t\bar{t}$ + jets). Full Run 2 analyses of $t\bar{t}H(b\bar{b})$ production have already been conducted by the ATLAS [7] and CMS [8] collaborations with signal strengths of 0.35 $^{+0.36}_{-0.34}$ and 0.33 ± 0.26, respectively. The recent ATLAS re-analysis superseding Ref. [7] implements several improvements in the calibrated objects, the systematics model, and an overhauled analysis strategy.

As this publication will summarise the analysis strategy and its results, details on many aspects, such as the background and systematics model, will not be outlined for brevity. A detailed account of the entire analysis is published as Ref. [3].

## 2  Analysis Strategy

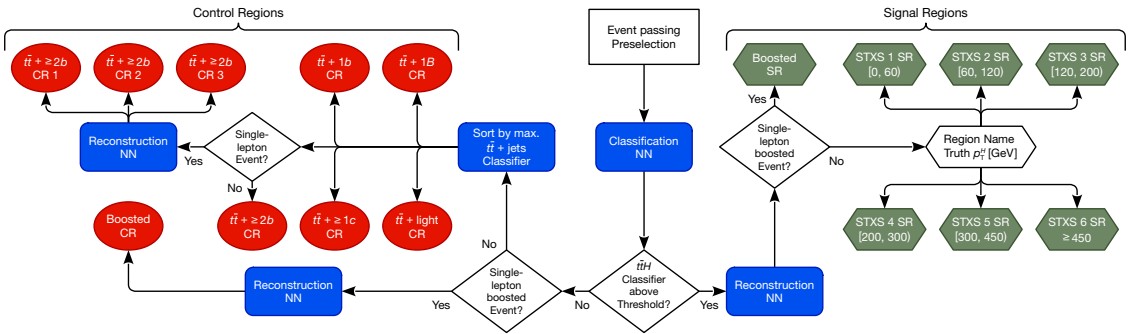

Figure 1: Flowchart depicting the multi-variate analysis setup used to assign events into signal and control regions via separate neural networks for event classification and Higgs boson reconstruction. The Figure is taken from Ref. [3].

Following dedicated studies on the dominant $t\bar{t}$ + jets background modelling [9], the background was split up into five separate categories based on the flavour of the jets not originating from top quark decays: $t\bar{t}$ + light and $t\bar{t}$ + ≥ 1$c$ only contain additional light and $c$-jets, respectively. Meanwhile, $t\bar{t}$ + ≥ 1$b$ is split into $t\bar{t}$ + ≥ 2$b$, $t\bar{t}$ + 1$B$, and $t\bar{t}$ + 1$b$ components corresponding to at least two additional $b$-jets, one additional jet containing at least two collimated $b$-hadrons, and one additional jet containing exactly one $b$-hadron, respectively. Unlike the first full Run 2 analysis, the region definition is entirely based on multi-variate discriminants to allow constraining each $t\bar{t}$ + jets category via a normalisation factor using dedicated control regions as pictured in Figure 1. The $t\bar{t}H$ signal region is additionally split by $p_{\mathrm{T}}^{H}$, the Higgs boson transverse momentum, for a differential measurement in the Simplified Template Cross-Section (STXS) formalism [6]. Details on the neural networks used for event categorisation and Higgs boson reconstruction are deferred to the next section.

Thanks to the classification-based region definitions following from the powerful categorisation scheme, the kinematic preselection could be loosened compared to the first full Run 2 analysis, leading to an improvement of the $t\bar{t}H(b\bar{b})$ signal acceptance by a factor of three: events in the single-lepton and dilepton top quark pair decay channels are targeted via kinematic preselections of ≥ 5 jets in association with exactly one lepton (electron or muon), and ≥ 3 jets in association with exactly two leptons, respectively. Because of the $b$-quarks expected from the decays of the top quarks and the Higgs boson, ≥ 3 $b$-tags are required in each channel. Additionally, single-lepton events in which both $b$-quarks from the Higgs boson are contained in a single $\Delta R = 1.0$ jet are selected in a separate boosted channel designed to increase sensitivity at high $p_{\mathrm{T}}^{H}$. Simultaneously, these events are vetoed in the single-lepton resolved selection.

## 3 Transformer Network Performance

The simultaneous measurement of the individual $t\bar{t}$+jets normalisation factors and the $t\bar{t}H$ signal cross-section is primarily possible because of the sophisticated multi-variate analysis strategy used for event classification and Higgs boson reconstruction. The transformer architecture [10] underlying the neural networks used for classification and reconstruction was originally presented in the context of natural language processing. It is agnostic to the cardinality of input objects and invariant under their permutation. These properties render it especially useful for tasks involving high-energy particle collision events, in which the number of final state particles is variable and where no unambiguous meaningful ordering exists between final state objects. The event classification and Higgs boson reconstruction networks use low-level four-vector and $b$-tagging information of all jets, leptons, and the missing transverse energy as input variables.

The classification networks predict the most probable event type from among the $t\bar{t}H$ signal and the five $t\bar{t}$+jets categories introduced in Section 2. In order to use the full information of the multi-class classification probabilities $p_i$ and account for differences between the process yields $N_i$, the classifiers are converted into six class discriminants $d_i$ as

$$d_i = \frac{p_i}{\sum_{j\neq i} p_j \hat{N}_{ij}}, \qquad \text{where} \quad \hat{N}_{ij} = \frac{N_j}{\sum_{k\neq i} N_k}, \tag{1}$$

corresponding to a ratio of the classifier of the target class $i$ and the average classifier of all other classes $j$ weighted by the corresponding process yield fraction $\hat{N}_{ij}$. In each channel, events are assigned to the signal region based on a $t\bar{t}H$ discriminant threshold optimised to maximise the signal-over-noise ratio $S/\sqrt{B}$. Events not passing this threshold are instead assigned to the control region corresponding to the highest $t\bar{t}$+jets discriminant value. As the discriminants defined in Equation 1 still show good separation power after the region assignments, the discriminant of each region's corresponding target process was used as the fit variable in the binned profile likelihood fits performed to extract the inclusive and differential $t\bar{t}H$ cross-sections.

The reconstruction networks use a pairing layer operating similarly to the tensor attention layer of SPANet [11] to infer the two most likely jets originating from the Higgs boson decay. The four-vectors of these jets are then combined to reconstruct $p_T^H$ for the differential cross-section measurement. The resulting confusion matrices between the true and the reconstructed $p_T^H$ for the signal events falling in the signal regions are shown in Figure 2. In the boosted channel, the reconstructed $p_T^H$ is used as the fit variable instead of the discriminants used in the other channels.

## 4 Analysis Results

Good post-fit agreement was observed after a simultaneous binned profile likelihood fit in all regions. Two fit setups were employed to measure both an inclusive $t\bar{t}H$ signal cross-section and a differential $p_T^H$ cross-section in the six STXS bins. The inclusive best-fit cross-section of

$$\sigma_{t\bar{t}H}^{\text{obs}} = 411\,^{+101}_{-\,92}\,\text{fb} = 411 \pm 54\,(\text{stat.})\,^{+85}_{-75}\,(\text{syst.})\,\text{fb}$$

is compatible with the Standard Model prediction of $\sigma_{t\bar{t}H}^{\text{SM}} = 507\,^{+35}_{-50}\,\text{fb}$ [6] at a Higgs boson mass of $m_H = 125.09\,\text{GeV}$. The observed event excess corresponds with a significance of $4.6\,\sigma$ over the background-only hypothesis. The differential cross-section results are shown in Figure 3 with correlation coefficients not exceeding 30 % and compatibility with the Standard

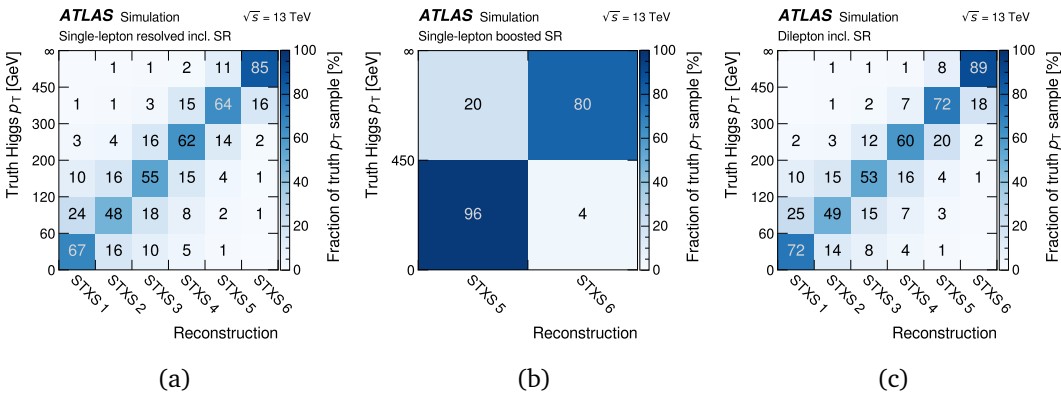

Figure 2: Confusion matrices of the Higgs boson transverse momentum ($p_T^H$) reconstruction in the signal regions of (a) the single-lepton resolved, (b) the single-lepton boosted, and (c) the dilepton channel. For each truth $p_T^H$ range, the fraction of events assigned to each signal region $p_T^H$ bin is given. The Figures are taken from Ref. [3].

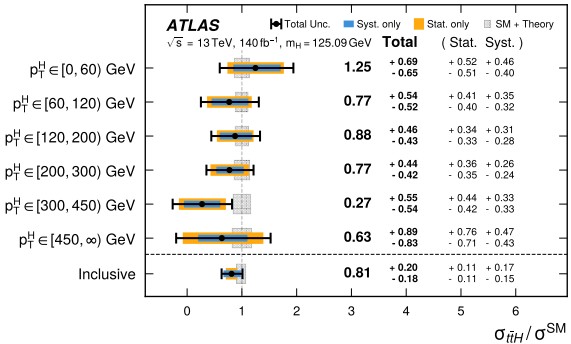

Figure 3: Measurements of the $t\bar{t}H$ cross-section in bins of the truth Higgs boson $p_T$ and inclusively. The Higgs boson rapidity is restricted to $|y_H| \leq 2.5$ for the differential measurement in line with the STXS prescription. Separate uncertainties for measurement and prediction are included. The Figure is taken from Ref. [3].

Model prediction at a p-value of 89 %. Modelling of the $t\bar{t} + \geq 1b$ backgrounds does not constitute the dominant uncertainties, unlike in the first full Run 2 ATLAS analysis. Instead, the $t\bar{t}H$ signal modelling represents the leading sources of systematic uncertainty.

## 5 Conclusion

A cross-section measurement of Higgs boson associated top quark pair production in the bottom–anti-bottom Higgs boson decay channel using the entire Run 2 proton–proton dataset collected at 13 TeV with the ATLAS detector at the Large Hadron Collider between 2015 and 2018 was presented. It represents a re-analysis of the data incorporating various improvements to the object and systematics model and the analysis strategy. Among these, the utilisation of sophisticated state-of-the-art transformer neural networks for the multi-variate analysis strategy allowed simultaneous measurements of the dominant $t\bar{t} + $ jets background process split into five disjoint categories and the signal process. Additionally, the good event classification performance made it possible to loosen the analysis preselection and increase the signal acceptance by a factor of three compared to the first full Run 2 analysis. The measured signal

cross-section of $\sigma_{t\bar{t}H}^{\text{obs}} = 411 \, ^{+101}_{-\,92}$ fb is compatible with the Standard Model prediction and represents the most precise single-channel $t\bar{t}H$ cross-section measurement to date. An excess of $4.6\,\sigma$ over the background-only hypothesis was observed.

## Acknowledgements

The author would like to acknowledge funding through the SNSF project grant 200020_212127 called "At the two upgrade frontiers: machine learning and the ITk Pixel detector".

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
