# Peer review of "Transformer Neural Networks in the Measurement of $t\bar{t}H$ Production in the $H\,{\to}\,b\bar{b}$ Decay Channel with ATLAS"

_SciPost Physics Proceedings_

## Round 1 · Referee Report · Anonymous (Referee 1) · 2024-12-26

Strengths

  1. The paper describes a measurement of the ttH cross section in the Higgs to bb decay channel. A ML technique is used to classify events, contributing to an improvement in background handling and resulting in a significant improvement in sensitivity over a previous analysis with the same data.
  2. These proceedings present a clear summary of the analysis strategy and results, as presented at the TOP2024 conference.

Weaknesses

  1. There are some minor areas where the text and clarity could be improved.

Report

The paper is a clear summary of a recent measurement of ttH process by the ATLAS experiment. I would recommend it to be published.

Requested changes

  1. [Abstract] Suggest quoting both expected and observed significance

  2. [Introduction section] Regarding the sentence "ttH production has been...": It is good to avoid starting sentences with symbols, so perhaps the sentence could be reworded to something like "The ttH production process has been..."

  3. [Introduction section] Regarding the sentence "The recent ATLAS re-analysis superseding..." and the sentence following it: Does this refer to the Ref. 3 analysis? It might be useful to state that more explicitly in that sentence. And, in the following sentence, it might be useful to clarify more explicitly how this publication relates to Ref. [3] (e.g., "This publication will summarize the analysis strategy and results of the Ref. [3] analysis...").

  4. [Analysis strategy section] Regarding the naming of the ttbar+X categories: It would be good to clarify if "b" refers to b-tagged jets and "B" refers to B hadrons. The text states that "tt+1B" corresponds to events with "one additional jet containing at least two collimated b-hadrons", but "tt+1b" corresponds to "one additional jet containing exactly one b-hadron". It would seem it might be more clear if the former would be referred to as "tt+2B" and the latter "tt+1B", so it would be good if this could be double checked or clarified.

  5. [Analysis strategy section] "..to allow constraining each tt+jets category..": This could be reworded to "to allow each tt+jets category to be constrained.."

  6. [Analysis strategy section] "Details on the neural networks used for event categorisation and Higgs boson reconstruction are deferred to the next section.": Please link to the specific section (or refer to it by name).

  7. [Analysis strategy section] "Thanks to the classification..": Suggest that the word "Thanks to" could be replaced with something like "Because of"

  8. [Analysis strategy section] As a suggestion on the last two paragraphs of this section, it may be useful to specify a bit more explicitly (or at least earlier in the paragraphs) which of these categories are control regions and which are the signal regions, so that as the reader can more easily map the text to the flowchart in Figure 1.

  9. [Results section] It seems lower case sigma is being used for cross section and for significance. For the significance, I suggest writing "standard deviation" instead of using sigma.

  10. [Results section] Please also include the expected significance (not just observed).

  11. [Results section] It may also be useful to mention how this measurement fits into the landscape of measurements in this sector (e.g., if it is the most precise measurement of this process in a single decay channel, it may be useful to state this).

Recommendation

Publish (meets expectations and criteria for this Journal)

  • validity: high
  • significance: good
  • originality: good
  • clarity: good
  • formatting: good
  • grammar: good

Author:  Chris Scheulen  on 2025-01-13  [id 5117]

(in reply to Report 1 on 2024-12-26)
Category:
remark
question
answer to question

Dear Referee,

Thank you very much for giving the submitted proceeding a read and suggesting improvements. I will respond to the individual points in line.

  1. [Abstract] Suggest quoting both expected and observed significance

Originally, the abstract was held short in order not to exceed the page limit recommendation by too large a margin. However, adding a sentence at the end to quote observed and expected significances sounds like a worthwhile addition.

  1. [Introduction section] Regarding the sentence "ttH production has been...": It is good to avoid starting sentences with symbols, so perhaps the sentence could be reworded to something like "The ttH production process has been…"

Good point, happy to change the sentence here if taken up in the recommendations by the editor-in-chief.

  1. [Introduction section] Regarding the sentence "The recent ATLAS re-analysis superseding..." and the sentence following it: Does this refer to the Ref. 3 analysis? It might be useful to state that more explicitly in that sentence. And, in the following sentence, it might be useful to clarify more explicitly how this publication relates to Ref. [3] (e.g., "This publication will summarize the analysis strategy and results of the Ref. [3] analysis...").

Indeed, the recent re-analysis is the one contained in Ref. [3]. I’m happy to add the reference explicitly here again and would suggest the following change: “The recent ATLAS re-analysis published as Ref. [3] supersedes Ref. [7]. It implements several […]”. Regarding the comment on the following sentence, I am not sure I fully understand. Currently, this paragraph outlines the scope of the proceeding, while referring to Ref [3] for the full analysis details. If the suggestion is to shorten this slightly, I am not opposed. For instance, “This publication will focus on analysis strategy and results of the re-analysis published as Ref [3] in full detail.”, might be fitting.

  1. [Analysis strategy section] Regarding the naming of the ttbar+X categories: It would be good to clarify if "b" refers to b-tagged jets and "B" refers to B hadrons. The text states that "tt+1B" corresponds to events with "one additional jet containing at least two collimated b-hadrons", but "tt+1b" corresponds to "one additional jet containing exactly one b-hadron". It would seem it might be more clear if the former would be referred to as "tt+2B" and the latter "tt+1B", so it would be good if this could be double checked or clarified.

The nomenclature here directly follows that used by ATLAS for the full Run 2 ttH legacy analysis of Ref. [3], in which ‘b’ refers to a single b-hadron matched to a jet, while ‘B’ refers to multiple b-hadrons matched to the same jet. If possible, I would like to retain the naming scheme to avoid confusion between this proceeding and the full analysis account in Ref. [3], especially since there already is some discrepancy between the ATLAS and CMS nomenclature for the ttbb categories.

  1. [Analysis strategy section] "..to allow constraining each tt+jets category..": This could be reworded to "to allow each tt+jets category to be constrained.."

That would help with readability, happy to implement this suggestion.

  1. [Analysis strategy section] "Details on the neural networks used for event categorisation and Higgs boson reconstruction are deferred to the next section.": Please link to the specific section (or refer to it by name).

Happy to change this sentence to “[…] are deferred to Section 3.” using a cross-reference.

  1. [Analysis strategy section] "Thanks to the classification..": Suggest that the word "Thanks to" could be replaced with something like "Because of"

Good point, happy to implement the change to “Because of”.

  1. [Analysis strategy section] As a suggestion on the last two paragraphs of this section, it may be useful to specify a bit more explicitly (or at least earlier in the paragraphs) which of these categories are control regions and which are the signal regions, so that as the reader can more easily map the text to the flowchart in Figure 1.

I am not sure I fully understand this request. Currently, the assignment of the tt+jets categories into dedicated control regions and that of the remaining ttH category into signal regions separated by Higgs boson reco pT is outlined in the second half of the first paragraph. I do not believe that it would be useful to describe this assignment before introducing the tt+jets background categories, especially given the importance of the independent background normalisation factors in the overall analysis. Perhaps, it could be added early in the paragraph that dedicated control regions were drawn for the tt+jets background components, so that the reader can appreciate the link between the regions and the categorisation: “[…], the background was split up into five separate categories with dedicated control regions based on the flavour of the jets not originating from top quark decays: […]”.

  1. [Results section] It seems lower case sigma is being used for cross section and for significance. For the significance, I suggest writing "standard deviation" instead of using sigma.

Good shout, happy to implement this change to reduce confusion between significance and cross-section.

  1. [Results section] Please also include the expected significance (not just observed).

Happy to quote both values here and in the conclusion.

  1. [Results section] It may also be useful to mention how this measurement fits into the landscape of measurements in this sector (e.g., if it is the most precise measurement of this process in a single decay channel, it may be useful to state this).

Currently, this is performed in the conclusion. To reduce the overall length of the proceeding, this is not explicitly repeated in the results section. I would therefore not want to duplicate the statement unless it is felt strongly that it should also be mentioned here.

Best Wishes,

Chris

Anonymous on 2025-01-18  [id 5135]

(in reply to Chris Scheulen on 2025-01-13 [id 5117])
Category:
remark

Dear Chris,

Thank you for addressing all of my comments. Regarding the Suggested Change #8, I apologize that my suggestion was unclear, but your proposed addition to the paragraph sounds like it would successfully help to address my original concern. All of your other answers and proposed changes also sound very good to me.

Best wishes

---

## Editorial Decision

accepted_in_target_journal